# Determination of Performance of No-Till Seeder and Stubble Cutting Prototype

Mehmet Emin Bilgili [1] , Yasemin Vurarak [1,*] and Ali Aybek [2]

1 Eastern Mediterranean Agricultural Research Institute Directorate, 01375 Adana, Turkey
2 Biosystems Engineering Department, Faculty of Agriculture, Kahramanmaraş Sütçü İmam University, 46000 Kahramanmaraş, Turkey
* Correspondence: yasemin.vurarak@tarimorman.gov.tr

**Abstract:** One of the most common problems in maize production is the management of plant residues. Small agricultural enterprises, which cannot allocate capital for acquiring stalk cutting machines for their operation, face many technical problems in preparing the sowing bed for the products that will be planted after maize. Stalks of maize that cannot be shredded adequately and on time cause machinery to clog, prevent the preparation of a proper sowing bed, increase fuel consumption and increase costs. The aim of this study is to compare the no-till sowing machine prototype and stalk cutting machine prototype with the classical stalk cutter in terms of some management values. The prototype stubble cutting machine used in the study was manufactured with a cylindrical structure and equipped with 24 cutting blades 1 cm thick and 8 cm wide. İn addition, the prototype stubble cutting machine used in the study was manufactured with a cylindrical structure and equipped with 24 cutting blades 1 cm thick and 8 cm wide. İn addition, the no-till seeder prototype was manufactured as a bucket-type seed hopper equipped with granular fertilizer capable of sowing four rows. It was concluded that the stubble cutting machine prototypes resulted in less fuel consumption with lower penetration resistance when compared with the classical stalk shredder.

**Keywords:** stubble management; no-till sowing; stalk cutting; post-harvest; prototype

## 1. Introduction

The Çukurova region in the south of Turkey is comprised of plains with one of the country's most fertile agricultural lands and good soil structure suitable for irrigated agriculture and production potential. Irrigated agriculture with high production potential is common in the region, and it is possible to grow 2–3 crops per year due to the Mediterranean climate. The superior advantages provided by the ecology have allowed the cultivation of many field crops in the Çukurova region, and one of these plants is maize (*Zea mays* L.). However, as reported in the survey study examining the harvesting and threshing traditions of Çukurova farmers, the farmers burn all the remaining maize stalks in the field (stubble burning) after harvest, which is a well-known problem waiting for a solution [1]. In addition to the low organic matter content of the soil in the region [2], the traditional high number of tillage operations and illegal burning of stubble residues [3] are among the leading problems that limit the sustainability of soil productivity. Designing and manufacturing agricultural machinery suitable for these activities that help sustainable management of agricultural lands are extremely important for the agricultural future of the region. Soil resources in Turkey as well as in the world are limited and cannot be increased. Reckless use of soil resources is one of the most important factors that increases the severity of the problem [4]. In addition, from an ecological point of view, it can be said that millions of tons of plant waste such as stubble, straw and paper raw materials are being burned [5]. A study carried out in the Çukurova region revealed that stubble is an important source of organic matter, and that 93–98% of stubble buried at a depth of 12 cm is transformed

into humus within 18 months. The decomposition rate of the stubble left on the surface, however, was approximately 1/3 of the buried amount [6].

In order to cope with an increase in precipitation in the autumn grain sowing season in recent years, producers have burned plant residues left after the harvest of maize to avoid delaying the sowing time, causing the deterioration of soil, air and water resources over time. The existing technology should be changed and no-till seeders and stalk cutters should be included in the machine parks to ensure that stubble is collected without burning and some of it is mixed with the soil. With the help of the pressure of local governments and non-governmental organizations, and the subsidies granted for the machinery by the government to prevent stubble burning, a decrease in burning maize stubble has ocurred in the region, albeit partially. However, the amount of capital allocated to these machines and the tractor power requirement for their use limit the purchase demand of small producers. The no-till seeders and stalk cutters on the market mainly consist of heavy tonnage machines that require high tractor power. This situation has created a driving force for researchers to design prototype machine sets with low costs, which can be an alternative to existing machine sets and can perform no-till seeding and stalk cutting work as required. The region is located in the Çukurova region, which is Turkey's largest plain, and the airport is frequently closed to flights due to stubble burning. There is an urgent need for machines that farmers can easily buy to prevent stubble formation. A conventional tillage seeder and stubble mower costs around $15,000 on average and is very expensive for our farmers. The prototypes cost about $2000.

Using no-till machines in the sowing system is now also a necessity for maintaining soil fertility in terms of global warming and climate change. However, it is known that access to machines used in this context is limited for small farm operations. The aim of this study is to present no-till seeder and stubble cutter prototypes to maize producers that are affordable for small enterprises. Another aim is to increase the farmers' options for preparing more economical seedbeds without burning stubble.

## 2. Materials and Methods

### 2.1. The Trial Area and Trial Materials

In the study area, the Çukurova region, there is no problem in terms of soil depth, except for the coastal part. This region is slightly different from the typical Mediterranean climate. An evaluation of climatic data for many reveals that the long annual precipitation average is 647.5 mm and the annual average temperature is 18.74 °C [7]. The study was carried out for 2 years (2018, 2019 years) in the "36° 51′ 18″ N and 35° 20′ 51″ E" coordinated trial fields of the Eastern Mediterranean Agricultural Research Institute Directorate. The general soil properties at the station where the trial was carried out were determined as follows. The lime status of the soil is between 16 and 21%, organic matter amount is between 1.27% and 2.33%, cation exchange capacity ratios are between 21.24 and 38.02 cmol kg$^{-1}$, soil pH value is between 7.50 and 7.99 and P$_2$O$_5$ values are between 16 and 179 kg ha$^{-1}$. In addition, the silt ratio of the soil has been determined as between 25.3 and 53.7%, the sand ratio is between 9.7 and 51.6% and the clay ratio is between 23.1 and 41.8% [8]. The soil surface in the trial areas is stone-free and the slope is in the range of 0–1°. It is in the alluvial soil series in terms of soil structure. Soil texture is homogeneous.

Some data of the trial area are presented in Table 1, and some climatic parameters recorded during the trial are presented in Table 2.

**Table 1.** Soil properties of the trial area.

| Year | Depth (cm) | Saturation (%) | E.C. (dS m$^{-1}$) | pH | Lime (%) | Organic Matter (%) | Plant Availability (kg ha$^{-1}$) | | Physical Structure (%) | | |
|------|------------|----------------|--------------------|----|----------|---------------------|------------------|------------------|------|------|------|
| | | | | | | | P$_2$O$_5$ | K$_2$O | Sand | Silt | Clay |
| 2018 | 0–30 | 55.50 | 1.05 | 7.60 | 13.43 | 1.28 | 56.0 | 901.5 | 32.4 | 35.5 | 31.5 |
| 2019 | 0–30 | 62.7 | 0.295 | 7.75 | 16.42 | 1.63 | 121.0 | 577.0 | 31.4 | 37.7 | 30.9 |

**Table 2.** Some of the climatic data recorded by years during the trial.

| Years | Parameters | I | II | III | IV | V | VI | VII | VIII | IX | X | XI | XII |
|-------|-----------|-----|-----|-----|-----|-----|-----|-----|-----|-----|-----|-----|-----|
| 2018 | Temperature (°C) | 8.7 | 13.9 | 15.7 | 20.5 | 21.7 | 27.3 | 29.5 | 29.9 | 26.1 | 23.1 | 15.5 | 9.0 |
| 2019 | | 8.7 | 10.9 | 15.2 | 18.6 | 21.7 | 26.3 | 30.4 | 29.9 | 27.7 | 22.1 | 15.8 | 12.6 |
| 2018 | Relative Humidity (%) | 61.3 | 67.7 | 60.7 | 57.8 | 68.1 | 63.8 | 67.5 | 67.4 | 59.9 | 56.5 | 52.4 | 64.8 |
| 2019 | | 61.9 | 50.7 | 63.5 | 60.7 | 69.1 | 69.2 | 64.4 | 67.3 | 66.1 | 53.5 | 66.3 | 74.4 |
| 2018 | Precipitation (mm) | 24.5 | 9.7 | 12 | 1.8 | 12.7 | 18.4 | 0.2 | 8.2 | 24.3 | 3.2 | 22 | 194.8 |
| 2019 | | 49.6 | 0.6 | 65.6 | 62.4 | 51 | 10.6 | 0 | 0 | 11.2 | 40.6 | 119 | 49 |

| | | Sowing, harvest and stubble cutting dates during trial | | |
|---|---|---|---|---|
| 2018 | Sowing | 03.05.2018 | | |
| | Harvest | | 01.09.2018 | |
| | Stubble cutting | | 09.09.2018 | |
| 2019 | Sowing | 12.03.2019 | | |
| | Harvest | | 05.09.2019 | |
| | Stubble cutting | | 12.09.2019 | |

The technical specifications of all tools and machines used during the trial are given in Table 3.

**Table 3.** Technical specifications of agricultural tools and machines used in the trial.

| Equipment | Technical Specifications | | |
|-----------|--------------------------|---|---|
| | Working Width (cm) | No. of Units (pcs) | Working Depth (cm) |
| Goble | 280 | 18 | 15 |
| Chisel | 260 | 9 | 25 |
| Disk harrow | 220 | 22 | 15 |
| Back double blade plough | 210 | 5 | 20 |
| Back plug roller | 280 | 4 | 20 |
| Hoeing machine | 280 | 3 | 15 |
| Band fertilizer machine | 280 | 5 | 15 |
| Spraying machine | 1200 | 1 | - |
| Round baler | 140 | 1 | - |
| No-till seeder (pneumatic) | 280 | 4 | 5–8 |
| Stubble shredder | 400 | 36 | - |

The weights of the no-till seeder and stubble shredder machines, the technical specifications of which are given in Table 3, are 1100 kg and 1740 kg, respectively. The stubble shredder takes action from the PTO with 540 rpm (Figure 1).

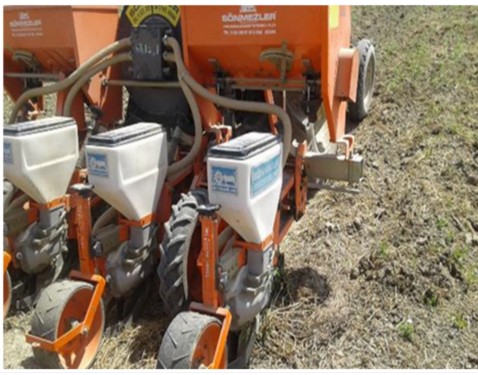
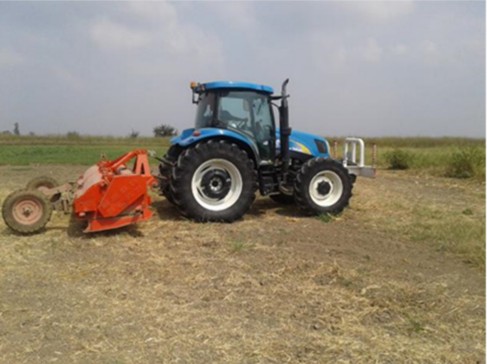

**Figure 1.** No-till seeder and stubble shredder used in the trial (traditional).

The technical specifications of the no-till seeder prototype and stubble cutter prototype are given in Table 4.

**Table 4.** The technical specifications of the no-till seeder prototype and stubble cutter prototype used in the trial.

| Technical Specifications | Stubble Cutter Machine Prototype | No-Till Seeder Prototype (Mechanic) |
| --- | --- | --- |
| Working width road condition (cm) | 280 | 300 |
| Working width (cm) | 240 | 280 |
| No. of blades/units | 24 blades | 4 units |
| Working depth (cm) | 5 | 5–8 |
| Weight (kg) | Empty weight: 1200 Full weight: 1450 | Empty weight: 775 |

Working principles of the prototype machines:

Stubble cutter machine prototype: The machine consists of assembling a cylinder with a working width of 240 cm and a diameter of 80 cm with the chassis. The top of the cylinder is equipped with a 240 cm blade section, parallel to the axis, with 24 blades which are 1 cm thick and 8 cm wide. The material of the cutter blades is steel and the distance between the two blades is 11 cm. They are joined to the cylinder by longitudinal welding. The cylinder of the stalk shredder, which has four pressure pallets on the front, can be used full or empty. The empty weight is approximately 1200 kg, and the full weight (with waste engine oil) is 1450 kg. Field and road positions are adjustable (Figures 2 and 3).

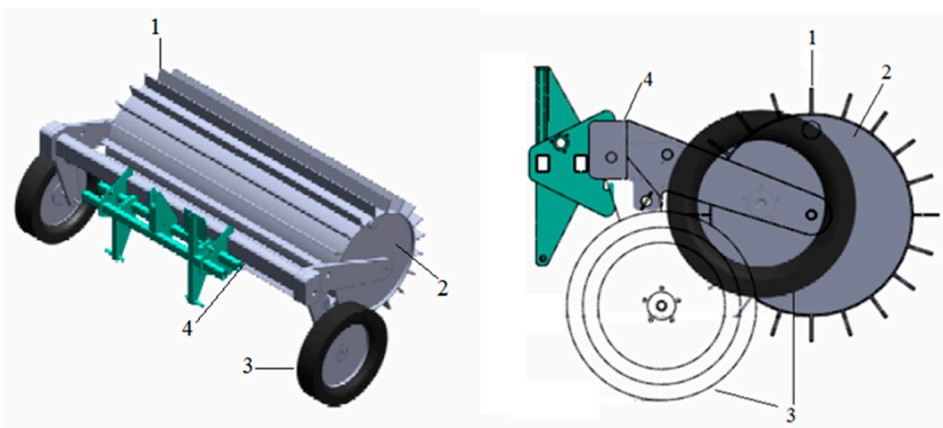

**Figure 2.** Design images of the stubble cutter prototype ((**1**) Stubble cutter blades; (**2**) Cylinder chassis; (**3**) Road condition wheel; (**4**) Main carrier chassis).

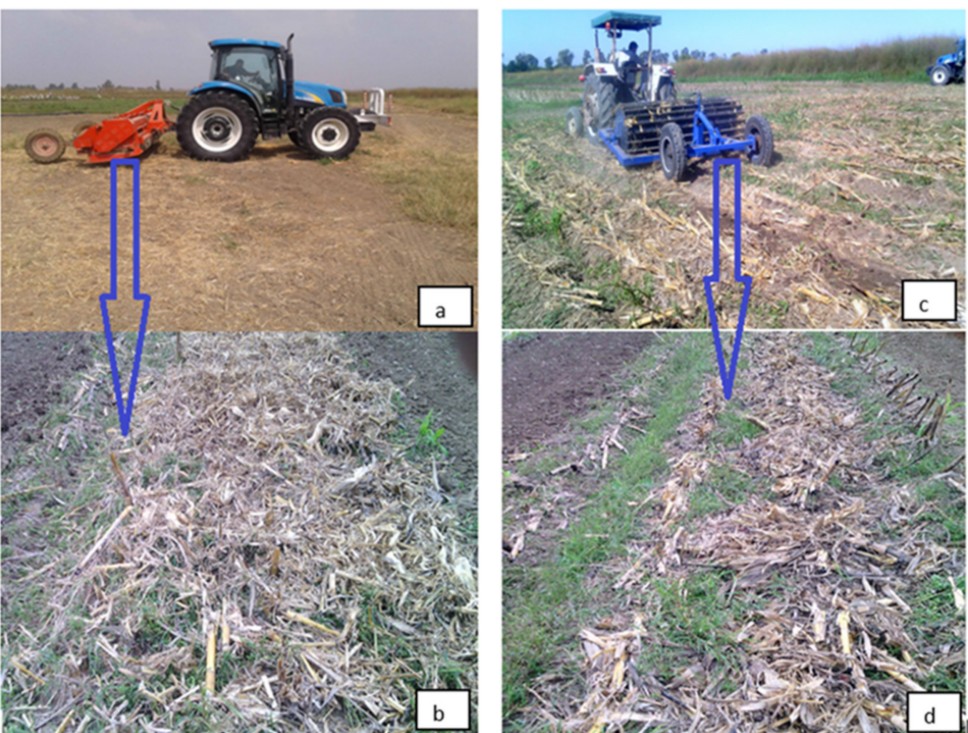

**Figure 3.** Recorded in experiments: (**a**) traditional stubble shredder; (**b**) efficiency of traditional stubble shredder; (**c**) prototype stubble cutter; (**d**) images of the effect of the prototype stubble cutter.

No-till seeder prototype: There is a spring-loaded stalk cutter disc in front of the machine and a 3 cm wide and 10 cm long soil cultivator rung for seedbed pre-preparation on the same axis behind the disc. These rungs are fitted with stubble cleaning flaps. After the seeds are released by the seeder, they are covered (pressure wheel). The elements that make up the whole machine are combined with a chassis and sowing can be completed in four rows. In addition, the seed hopper is designed with buckets, and there are two fertilizer hoppers in total, one for each sowing bucket, to fertilize with sowing (Figures 4 and 5).

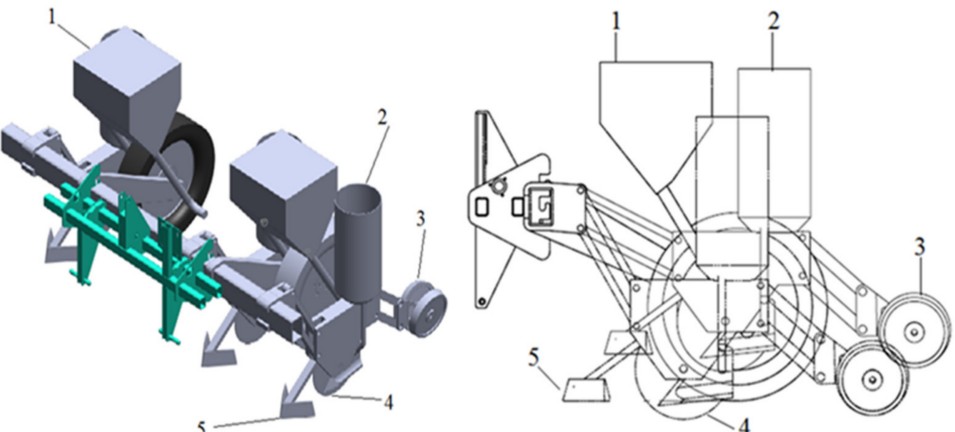

**Figure 4.** Design images of the no-till seeder prototype ((**1**) Fertilizer hopper; (**2**) Seed hopper; (**3**) Seed press roller; (**4**) Sowing coulter with cutting disc; (**5**) Row cleaning attachment).

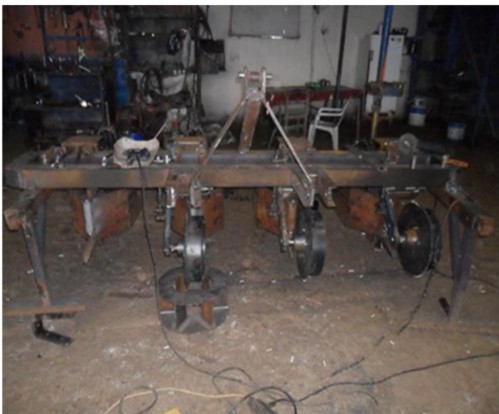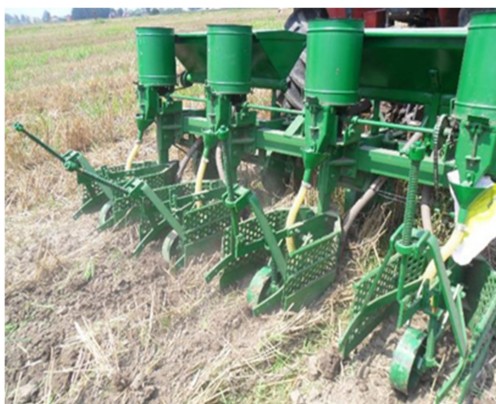

**Figure 5.** Construction and usage stages of the prototype sowing machine used in the research.

### 2.2. Agricultural Activities Carried out During the Trial

A registered maize variety planted in the plain was used as plant material in the trial. This variety is grain, P1429 corn. The FOA maturity group of the variety is GDD 1577 °C. The number of physiological maturation days is 115–117 days and its purity is 100%. Sowing was carried out at a depth of 4–6 cm with 70 cm between rows and intrarow of 18–20 cm, according to the subject (70 000–75 000 seeds ha$^{-1}$). Taking into account the deficiencies revealed by the soil analysis, 80 kg ha$^{-1}$ of $P_2O_5$ fertilizer and half of the nitrogen fertilizer (140 kg ha$^{-1}$) were applied to the strips during sowing, and the other half was applied before the first irrigation [9]. Trial plots comprised of 4 rows with a length of 55 m and a distance of 2.8 m between plots were prepared. Weed control was carried out and measures were taken to deal with maize stem worms, cob worms and aphids, as well as agricultural control for soybean in case of any disease or pest according to the relevant procedure [10]. In maize production, the first irrigation was performed approximately one month after the sowing, the second irrigation was completed when the tassel emergence was completed, the third irrigation took place when the cob emerged, the fourth irrigation was applied at grain formation and the fifth irrigation was carried out during the milk formation phases [11,12]. Maize leaves were harvested with a combine harvester when the husks dried, the cobs separated from the stem and the grains hardened (moisture 15%) [13].

### 2.3. Trial Subjects

Two no-till seeder and stalk shredder systems were examined comparatively within the scope of the study. Soil tillage and maintenance works in both subjects were conducted in the same way, at the same time, with the same machines. Accordingly, the mechanical systems that constitute the trial subjects are as follows:

System 1: No-till seeding method by stalk shredding (subject to control): The previous crop was shredded with a stalk shredder, collected and then the stubble was directly ridge planted with a no-till planter (traditional).

Machines used in System 1: Goble disc + ridge planter + no-till seeder + hoe + furrow plow with fertilizer + pulverizer + combine harvester + stalk shredder + cylindrical straw baling machine.

System 2: No-till sowing method by cutting the stubble (prototype): The plant stubble from the previous crop was shredded with the stubble cutter prototype, collected, then followed by ridge sowing with the no-till seeder prototype (alternative prototype).

Machines used in System 2: Goble disc + ridge planter + no-till seeder prototype + hoe + furrow plow with fertilizer + pulverizer + combine harvester + stubble cutter prototype + cylindrical straw baling machine.

The goble disc was used before sowing in both systems. This is because the Çukurova region is a region that receives a lot of precipitation during the winter months, and the weed problem is quite abundant and the farmers cannot cope with it. Soil tillage is performed 5 months before sowing the main crop of corn, which is planted on pre-prepared ridges

with a no-till seeder. For our region, the sowing beds must be prepared in advance in order to eliminate the weed problem before sowing the main crops.

### 2.4. Measurement, Analysis and Calculation Methods

References and methodologies of the measurement, analysis and calculation methods are given in Table 5. All measurements and calculations were made according to the methods reported in these references.

**Table 5.** Measurement method references.

| Measurement | References | Methodology |
| --- | --- | --- |
| Measurements related to machine performance | | |
| Stalk cutting stress, MPa | [14] | The maximum pressure applied by the blade to the plant stem has been calculated as an average, according to the method in the reference. |
| Machine labor requirement, h ha$^{-1}$ | [15,16] | The main usage times per unit of the machines in the system are measured with 3 replications from the starts to the end of the work and calculated to determine an average value as "ha" unit. The base time was taken into account in the measurements. |
| Human labor requirement, h ha$^{-1}$ | [15,16] | The main usage times per unit of the human labor in the system are measured with 3 replications from the starts to the end of the work and calculated to determine an average value as "ha" unit. The base time was taken into account in the measurements. |
| Tractor power requirement, kW | [17] | The tractor power required by the prototype machines was quoted from the reference guide according to operating width and weight. |
| Fuel consumption, L ha$^{-1}$ | [16–18] | Fuel consumption was measured in 3 repetitions per hectare for all machines in the system and calculated according to the tank filling method, and the average value was taken. |
| Measurements for plant growth | | |
| Number of plants, Unit per m$^{-2}$ | [13,19] | The number of plants per 1 m$^2$ on a row after emergence was measured with 3 replications, and their average was taken. |
| Plant height, cm | [13,19] | 10 randomly selected plants from each study plot were measured with a wooden meter in 3 replicates. |
| Stem thickness, cm | [20] | Measurements were made with a digital caliper for the first 5 cm of the stem level from the soil surface of the plant before harvesting 5 randomly selected plants from each plot in 5 repetitions, and the average value was taken. |
| Height of first cob from the ground, cm | [13,19] | Measurements were made in each system using a tape measure from the soil surface to the point where the first cob was tied in 5 randomly selected plants before harvest, and the averages were taken. |
| Stem moisture, % | [13,19] | 25 cm samples were taken from the stem of 5 randomly selected plants in each system before harvesting, and the average stem moisture was calculated by taking the difference between their weights before and after they were kept in an oven set at 105 °C for 24 h according to the dry base. |
| Stubble (Biomass) density and surface coverage rate, Kg m$^{-2}$ | [21–24] | A meter was placed on a 1 m × 1 m piece of wood material at an angle of 45° and markings were made at 1 cm intervals, and the points corresponding to the shredded stems were marked, counted, the total of the points in contact with the parts were divided by 100 cm and the surface coverage ratio was calculated by using the cross line method (%). Calculations were made with 3 replications for the systems and the averages were taken. This process was conducted separately for 2 different situations, before and after baling. |

**Table 5.** *Cont.*

| Measurement | References | Methodology |
|---|---|---|
| Biomass moisture, % | [13–19] | Samples weighing 500 g were taken from the stubble remaining on the soil surface after the harvest, with 3 repetitions, and the average stubble moisture was determined by taking the difference between their weights before and after they were kept in an oven set at 105 °C for 24 h on a dry basis. |
| Yield, t ha$^{-1}$ | [13,19] | For each system, the average grain moisture was expected to decrease to 15%, and cobs in 5 m$^2$ were harvested with 3 replications before harvesting and the averages were taken. |
| Measurements and analyzes related to soil structure | | |
| Bulk density, g m$^{-3}$ | [25] | Undisturbed soil samples at different depths with 3 replications were processed using 100 cm$^3$ core sets according to the cylindrical method, and the averages were taken to determine the soil volumetric mass. |
| Soil humidity, % | [26] | The average soil humidity was calculated by determining the difference in weight before and after the undisturbed soil samples were kept in an oven set at 105 °C for 24 h on a dry basis with 3 replications. |
| Soil penetration resistance, MPa | [27,28] | The averages were achieved by using a penetrelogger with memory, LSD display and GPS source that can measure in the range of 0–80 cm soil depth and 0–5 MPa to measure rows with 3 replications at different depths after. |
| Cost calculations | | |
| Input/output ratio | [15] | The input/output ratio was calculated by considering the annual records per unit area. Calculations were made as cost–benefit. |

At the end of the study that lasted for two years, the results of the measurements of the two systems were compared using the average, ratio and proportion methods, and the operating data of the machines in the alternative system were determined.

## 3. Results

The germination rate of the seed used was 90%, and the average field shoot emergence rate for 2 years according to the systems was 95% for system 1 and 86% for system 2, respectively. Some vegetative development data of the plants in the main crop maize production are given in Table 6 according to the years and systems. When an evaluation is made according to the average data, it is possible to say that the efficiency in System 1 is less than System 2, albeit at a low rate. If an evaluation is made in general, there is no considerable difference between the parameters for the whole development.

**Table 6.** Data obtained from the maize plant according to years in the trial.

| Years | Subject | Plant Height (cm) | Stem Thickness (mm) | Number of Plants (per m$^{-2}$) | Height of First Cob from the Ground (cm) | Stem Humidity (%) | Yield (t ha$^{-1}$) |
|---|---|---|---|---|---|---|---|
| 2018 | System 1 | 233 | 23.5 | 5.5 | 102 | 17 | 14.81 |
| | System 2 | 232 | 24.1 | 5.4 | 109 | 20 | 16.02 |
| 2019 | System 1 | 243 | 23.9 | 4.5 | 103 | 18 | 15.17 |
| | System 2 | 240 | 24.5 | 4.0 | 104 | 21 | 15.35 |
| System 1 ave. | | 238 | 23.7 | 5.0 | 102.5 | 17.5 | 15.01 |
| System 2 ave. | | 236 | 24.3 | 4.7 | 106.5 | 20.5 | 15.68 |

The stubble coverage rates on the land after harvest are given as before and after baling in Table 7. The stubble coverage rate, which is over 95% after the maize harvest

every year, has been collected by baling in such a way that 24–25% stubble remains. On average, stubble moisture was determined to be between 20 and 25% during both years.

**Table 7.** Biomass status of maize stubble on the field surface.

| Subject | Biomass Status | Years | |
|---|---|---|---|
| | | **2018** | **2019** |
| System 1 | Coverage rate after harvest (%) | 96 | 95 |
| | Coverage rate after baling (%) | 25 | 27 |
| | Biomass moisture (%) | 22 | 23 |
| System 2 | Coverage rate after harvest (%) | 97 | 96 |
| | Coverage rate after baling (%) | 24 | 25 |
| | Biomass moisture (%) | 20 | 21 |

The volume weight and soil moisture measurements determined in the plots where the systems were tested are given in Table 8. Considering the average soil volume weight and soil moisture values of both subjects, it is seen that there is no significant effect in terms of these parameters.

**Table 8.** The volume weights and moisture of the soils taken from the trial area of the systems.

| Year | Subject | Bulk Density (g m$^{-3}$) | | Soil Moisture (%) | |
|---|---|---|---|---|---|
| | | **0–15 cm** | **15–30 cm** | **0–15 cm** | **15–30 cm** |
| 2018 | System 1 | 1.68 | 1.69 | 15.94 | 16.14 |
| | System 2 | 1.63 | 1.71 | 15.34 | 15.73 |
| 2019 | System 1 | 1.69 | 1.70 | 15.97 | 16.06 |
| | System 2 | 1.64 | 1.72 | 15.35 | 15.72 |
| System 1 ave. | | 1.68 | 1.69 | 15.95 | 16.10 |
| System 2 ave. | | 1.63 | 1.71 | 15.34 | 15.72 |

Soil penetration resistance between 0 and 30 cm layers is given in Table 9. The soil penetration resistance values obtained in the current study were lower than the threshold of penetration resistance for field crops.

**Table 9.** Penetration resistances of the soils taken from the trial area of the systems.

| Year | Subject | Penetration Resistance (MPa) | | | | | |
|---|---|---|---|---|---|---|---|
| | | **0–5 cm** | **5–10 cm** | **10–15 cm** | **15–20 cm** | **20–25 cm** | **25–30 cm** |
| 2018 | System 1 | 0.65 | 1.25 | 1.37 | 1.40 | 1.39 | 1.53 |
| | System 2 | 0.52 | 0.98 | 1.07 | 1.26 | 1.41 | 1.63 |
| 2019 | System 1 | 0.58 | 1.23 | 1.35 | 1.43 | 1.43 | 1.50 |
| | System 2 | 0.48 | 0.95 | 1.05 | 1.20 | 1.40 | 1.50 |
| System 1 ave. | | 0.56 | 1.24 | 1.36 | 1.41 | 1.41 | 1.51 |
| System 2 ave. | | 0.50 | 0.96 | 1.07 | 1.23 | 1.40 | 1.56 |

The fuel consumption of the systems, the machine labor requirements and the income and expense ratios of the system are given in Table 10.

**Table 10.** Some operational data and cost status of the systems.

| Year | Subject | Machine Labor Requirement (h ha$^{-1}$) | Human Labor Requirement (h ha$^{-1}$) | Fuel Consumption (L ha$^{-1}$) | Input/Output Ratio |
|------|---------|------|------|------|------|
| 2018 | System 1 | 1.33 | 1.49 | 104.2 | 34.70 |
|      | System 2 | 1.51 | 1.69 | 95.4 | 40.85 |
| 2019 | System 1 | 1.26 | 1.41 | 103.8 | 27.25 |
|      | System 2 | 1.43 | 1.60 | 96.0 | 30.35 |
| System 1 ave. | | 1.29 | 1.45 | 104.0 | 30.97 |
| System 2 ave. | | 1.47 | 1.65 | 95.7 | 35.60 |

The total machine labor requirement values on the basis of systems was taken into consideration; the averages for 2 years, an average of 1.29 h ha$^{-1}$ for System 1 and System 2, are required, while this rate is 1.47 h ha$^{-1}$ for system 2 where a prototype machine is used. In addition, when the human labor requirements of the systems are examined, it has been determined that the labor requirement of system 1 is 1.45 h ha$^{-1}$ on average, and System 2 is 1.65 h ha$^{-1}$. In terms of fuel consumption, it has been determined that the average for 2 years is 104 L ha$^{-1}$ for system 1 and 95.7 L ha$^{-1}$ for system 2 where prototype machines are used. In terms of income/expense ratio, System 2 (35.60) was determined to be a more profitable investment than System 1 (30.97).

The fuel consumption and forward speeds of prototype machines and conventional machines on the market are given in Table 11. While the average fuel consumption of the prototype stubble cutter is 3.25 L ha$^{-1}$, the average fuel consumption of a conventional stalk shredder is 16.80 L ha$^{-1}$. It has been determined that the classical stalk shredder consumes 5.2 times more fuel than the prototype stubble cutter. In this respect, it is possible to say that the prototype stubble cutter will be effective in reducing costs. At the same time, it was determined that the speed of the prototype stubble cutter was slightly slower because it sunk into the soil while operating. During operation, the maximum pressure applied to the stubble cutting point was calculated as 0.81 MPa. The operating values of the prototype no-till seeder, on the other hand, consumed 9.5 L ha$^{-1}$, while the fuel consumption of the traditional no-till seeder was approximately 1/3 less. It has been determined that they are similar (4.4 km h$^{-1}$) in terms of forward speeds.

**Table 11.** Some operational parameters of prototype stalk cutters and classical stalk shredders.

| Equipment | Operating Width (cm) | Fuel Consumption (L ha$^{-1}$) | Forward Speed (km h$^{-1}$) | Necessary Tractor Power (kW) |
|-----------|------|------|------|------|
| Classic stalk shredder | 400 | 16.80 | 8.5 | 80 |
| Stubble cutter machine prototype | 280 | 3.25 | 7.8 | 37 |
| No-till seeder | 280 | 3.5 | 4.5 | 60 |
| No-till seeder prototype | 280 | 9.5 | 4.4 | 50 |

Possible malfunctions were also observed during the trials. During the two-year studies, no malfunctions occurred in the prototypes. However, in the no-till seeder prototype, there is a possibility of malfunctions in mechanical parts, chain and gear system over time. No malfunction is expected in the prototype stubble-cutting machine, except for the rolling bearing.

## 4. Discussion

Soil compaction is a factor that affects the germination rate of the crop to be planted. Whether in conventional or direct sowing, it is not desirable to have soil with a compaction level that affects germination. However, it is known that soil penetration resistance increases

as soil depth increases [28]. Looking at the 2-year averages of the systems, it has been determined that the penetrometer values in the range of 0–5 cm, 5–10 cm, 10–15 cm and 15–20 cm increase as the depth increases in System 1. That is, the mean penetration resistance measured for System 1 and System 2 at 0–5 cm depth was 0.56 MPa and 0.50 MPa, respectively, while this value was 1.51 MPa and 1.56 MPa at 25–30 cm depth, respectively. However, for both systems, there is no compression in the range that will restrict plant growth. It is thought that this can be associated with the total weights of the machines used in the systems. It is stated that the limit penetration resistance value, which prevents plant root growth, is 3 MPa [29]. It has been reported that this value is 3.6 MPa in soils where a conventional tillage system is applied and 5 MPa in soils without tillage [30]. If there are no permanent root canals and cracks in the soil, the critical value is considered to be 2 MPa [31]. In another study, it is reported that the 2 MPa limit value cannot be accepted as a limiting value for root growth for different tillage systems [32]. In addition, it is reported that in soils with high clay content, the critical limit value should be 2 MPa when a conventional tillage system is applied, 3 MPa when chisel cultivation is applied and 3.5 MPa when a direct sowing system is applied [33]. The soil penetration resistance values obtained in the current study were lower than the threshold limit of penetration resistance for field crops. It was suggested that the threshold of penetration resistance for field crops should be in the range of 2 to 3.5 MPa [34,35].

In terms of volume weight value, 1.68 g cm$^{-3}$ and 1.63 g cm$^{-3}$ are found for the traditional System 1 and alternative System 2, respectively, at a depth of 0–15 cm and 1.69 cm$^{-3}$ and 1.71 cm$^{-3}$, respectively, at a depth of 15–30 cm. As the depth increased, the volume weight increased slightly in both systems. However, there were no extreme differences between subjects. In a study, it was stated that the limit volume weight value for plant root development for clay loam soils above 1.40 g cm$^{-3}$ should be considered in terms of compaction potential [36,37]. It is stated that the ideal volume weight in clay loam soils is <1.40 g cm$^{-3}$, the volume weight that affects root growth is 1.63 g cm$^{-3}$ and the volume weight that inhibits root growth is >1.80 g cm$^{-3}$ [38]. In another study, it is stated that plants develop well when the soil volume weight is in the range of 1.15–1.45 g cm$^{-3}$, while a volume weight value greater than 1.55 g cm$^{-3}$ is not suitable for plant growth [39]. In a study conducted in Mexico, the effects of direct sowing and traditional tillage methods on the physical properties of a clay loam soil were investigated. The results showed that there was little difference between the tillage systems and the volume weight values were 1.21 g cm$^{-3}$–1.39 g cm$^{-3}$, respectively [40]. In a study investigating the effects of direct sowing and conventional tillage on soil properties and corn yield in southwest China, it was determined that the direct sowing method decreased soil volume weight by 7% in the upper 30 cm and increased soil moisture content by 3% compared to the conventional tillage method. In addition, they stated that corn yield increased by 11% in the direct sowing method compared to the traditional tillage method [41]. In our study, it was determined that the prototypes used in System 2 increased the efficiency by 4.2% compared to the other system. In parallel with our study, it has been stated that, at least in the short term, the direct sowing method may be a suitable alternative to reduce soil mobilization and its negative effects without any significant loss in crop yield [42].

The stubble coverage on the fields after harvest was measured twice, before and after baling. The stubble coverage rate, which was over 95% after the maize harvest during both years, is the main cause of stubble burning in the region. Maize stubble is mostly burned by the producers due to the lack of suitable equipment. A study revealed that a stubble yield of 7 t ha$^{-1}$ was reported, even for no-till plots where maize was planted directly as a second crop with an average yield was 8.2 t ha$^{-1}$, and this forced the producer in dealing with the most important problem in terms of stubble management [43]. In the same study, it was emphasized that it was necessary to work on different alternatives that would be easily accessible to producers to manage maize stubble after harvest.

The moisture of the soil and the humidity of the plant stubble or stems must be suitable to enable machines that are equipped to cut and process stubble to be able to do the cutting

work at a sufficient level. In many studies, it has been reported that if the soil surface is hard enough, the discs cut the stubble, and if it is soft, the stubble is buried into the soil. It has also been reported in the study that large diameter discs cut the stubble more easily [44,45]. Therefore, it is desirable that the surface soil moisture is slightly less to increase the performance of the prototype stubble cutter, as in System 2. In terms of trial results, it was an advantage that the humidity in the trial plots in System 2 was less than in System 1. It is thought that this may have had an impact on the prototype no-till seeder used in the system.

At the end of the study, the average total machine labor requirement on the basis of systems was 1.29 h ha$^{-1}$ and 1.47 h ha$^{-1}$ for System 1 and System 2, respectively. In addition, when the human labor requirements of the systems are examined, it has been determined that the labor requirement of System 1 is 1.45 h ha$^{-1}$ on average, and System 2 is 1.65 h ha$^{-1}$. In the system with prototype machines (System 1), the use of machine labor was found to be 12.3% higher. In a study conducted in Turkey, different sets were compared in terms of machine labor use. It was determined that 4.27 h ha$^{-1}$ machine labor is required for chisel + disc harrow + pneumatic sowing, 3.64 h ha$^{-1}$ for milling + pneumatic sowing set, and direct sowing set 2.44 h ha$^{-1}$ [46]. In this respect, it is possible to say that System 2, where our prototypes are located, has a very low machine labor requirement.

Fuel consumption was determined as 104 L ha$^{-1}$ and 95.7 L ha$^{-1}$, respectively. In terms of income/expenditure ratio, it was determined that System 2 (35.6) would be a more profitable investment than System 1 (30.97). In addition, some operating costs have been determined for prototype machines. It has been determined that the fuel consumption of the prototype stubble cutter is 3.25 L ha$^{-1}$, while a conventional stalk shredder consumes 16.80 L ha$^{-1}$ on average. Therefore, it has been determined that the conventional stalk shredder consumes 5.2 times more fuel than the prototype stubble cutter. The operating values of the prototype no-till seeder, on the other hand, consumed 9.5 L ha$^{-1}$, while the fuel consumption of the traditional no-till seeder was approximately 1/3 less. It is thought that the reason for this is that the no-till seeder prototype is mechanical, and it tends to dive deep from time to time and is exposed to excessive load. Researchers in different regions continue to work on different types of prototype stalk shredders according to the products grown in the regions and the soil structure. For example, in a study conducted in the Thrace Region of Turkey, a prototype stalk shredder was used together with a disc harrow and milling type shredders used by the producers to shred sunflower stalks. It has been determined that the most important indicator of economy in shredders is fuel consumption, which has been determined as 11.03 L ha$^{-1}$ for disc harrows, 14.47 L ha$^{-1}$ for milling type shredders, 6.74 L ha$^{-1}$ for prototype stalk shredders (suspended type) and 5.76 L ha$^{-1}$ for prototype stalk shredders (towed type) [47].

When the stubble coverage rate was examined, it was determined that the average data of 2 years in System 1 and System 2 subjects were 26% and 24.5%, respectively. In the study, it was determined that the corn stubble surface cover was 7%, 35% and 27%, respectively, after ear plough + two times disc harrow + sowing, chisel + disc harrow + sowing, rotary tiller + sowing and direct mulch sowing. In addition, it was reported as 39% in direct stubble sowing [48].

In general, the stalks do not lose contact with the soil during the shredding process performed with conventional type stalk shredders, which has an adverse effect on the performance of the sowing machines for subsequent plantings. While long stubble causes a decrease in the sowing depth and sowing uniformity, and seeds are more dispersed into the environment, at the same time, they are bent and buried deeper into the soil by the furrow opened compared to short stubble [49]. This was not encountered in the performance of the prototype stubble cutter, and the root part remaining in the soil was cut and fragmented by the vertical penetration of the machine's cutter blade.

Researchers suggest that we cannot think to save our soil if we keep treating it as a factory, sticking to a "fuel in–food out" model. Soil is not a machine; it is a living being of astonishing, and still largely unexplored, complexity. To seriously address the soil issue,

we should agree that it is necessary to rethink the structure and functioning of the whole farming system to start with [50]. Stubble management is the most important part of sustainable soil management. Alternative prototype machines will continue to increase in importance day by day, without burning the stubble.

Looking at international studies, it is seen that there are different designs on the stalk-cutting prototype machines. Some of these are prototypes that are mounted on the combine, while others are designed to be used by connecting to the tractor. However, it has been reported that stalk-cutters mounted on the combine reduce the combine's performance [51]. This also shows that stalk-cutters should be able to deal with different alternatives without being dependent on the combine. In many countries, such as Turkey, studies on no-tillage sowing have increased in recent years. In order to eliminate the straw coverage problem, there are many different stubble-sowing machine prototype studies. One of them is an essential prototype of no-tillage sowing made in southwest China. In this study, a no-tillage sowing machine with a bidirectional stubble-cutting apparatus was designed, and working parameters were determined in field conditions [52]. It is thought that increasing the amount of stubble cultivation area will be a success for every country by conducting more studies like this.

At the end of the study, it is thought that the stubble cutter recommended to the producers is likely to be used by the producers over time. However, it should be known that change will not be rapid, and habits cannot be abandoned quickly.

### 5. Conclusions

Only the traces of the cutter blades are visible on the field surface because of the rolling operation of the machine on the field surface. According to the 2-year data of the study, it was determined that the classical shredder used in the study consumed 5.2 times more fuel than the prototype stubble cutter. It can be said that this has a direct impact on the income/expense ratios and reduces costs. While the tractor power required for prototype machines is 37–60 kW, a more powerful (>60 kW) tractor is needed for the other machine. Some of the advantages of the prototype stubble cutter include simple production technology for small industrial enterprises, easy production and short maintenance and repair time in case of malfunction, and the availability of spare parts. This will also allow small businesses to expand their machine parks by reducing the acquisition capital of the machines. Furthermore, it can be said that especially the stubble cutter can be used for other purposes such as weed crushing/cutting and silage compaction. However, using the prototype no-till seeder can have a negative impact on costs since the fuel consumption is three times higher than the classic stalk shredder.

**Author Contributions:** Conceptualization, M.E.B., Y.V. and A.A.; methodology, M.E.B. and Y.V.; formal analysis, M.E.B., Y.V. and A.A.; writing—original draft preparation, Y.V. and M.E.B.; writing—review and editing, Y.V. and A.A.; supervision, M.E.B., Y.V. and A.A.; funding acquisition, M.E.B., Y.V. and A.A. All authors have read and agreed to the published version of the manuscript.

**Funding:** This project: number TAGEM/TSKAD/14/A13/P08/07, was funded by the Ministry of Agriculture and Forestry Eastern Mediterranean Agricultural Research Institute of Turkey.

**Institutional Review Board Statement:** Not applicable.

**Informed Consent Statement:** Informed consent was obtained from all subjects involved in the study.

**Data Availability Statement:** The data presented in this study are available on request from the authors.

**Conflicts of Interest:** The authors declare no conflict of interest.

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
