# Peer review of "Determination of Performance of No-Till Seeder and Stubble Cutting Prototype"

_agriculture, doi:10.3390/agriculture13020289_

Round 1
Reviewer 1 Report
The article deals with the issue of post-harvest treatment of agricultural land for the establishment of subsequent field culture. In the subject area on which the research of the authors of the article was focused, i.e. Çukurova Region in South of Türkiye, the characteristic method of this preparation is the widespread burning of post-harvest plant residues in the fields. The authors of the article in its introduction (chapter 1) quite rightly state that this method (i.e. burning waste biomass) is not suitable and that it would be much better for the soil if the remaining biomass (mainly maize stalks) disintegrated and remained on the surface of the soil to decomposition. This would, among other things, enrich the soil with humus. It is therefore a preference for no-till cultivation technologies, using specific no-till sowing machines and stalk cutters. Such machines are known, but in the given conditions they are unaffordable for ordinary farmers due to their high purchase price and operating costs. Therefore, the authors of the article focused their research on the creation and evaluation of machines that would meet the requirements of the no-till method of growing agricultural crops and at the same time be economically available for local farmers in the Çukurova Region. This intention of the authors is certainly very commendable, and I believe that the importance of this issue goes beyond the subject area of Çukurova Region, as similar problems are also being solved in many other areas of the world. I therefore consider the article appropriate and its material content beneficial for the scientific and practical sphere of agriculture.
Chapter 2 – Materials and Methods presents in subsection 2.1 both the natural conditions in the area where the field research work was carried out, as well as the basic characteristics of the machines used, including stubble cutter and no-till seeder prototypes. The description of the design parameters of both new prototypes is comprehensible and is appropriately supplemented with pictures, but I recommend specifying the principle of the seeding device used in the no-till seeder prototype (mechanical, pneumatic, etc.). I also recommend checking the correct use of the technical term "planting" in connection with no-till seeder. In my opinion, the term "planting" refers to the planting of plants, whereas with seeders we work with seeds, so the correct term is "sowing" (eg lines 118 or 129). I would also recommend that information be added on how long after the corn harvest experimental work with the disintegration of plant residues was carried out and how these plant residues (stalks) looked (e.g. whether they were green, dry, withered, what was the thickness of the stems, etc. ).
Subchapter 2.2 describes the agrotechnical parameters of the established experiments with no-till sowing of maize.
I consider subchapter 2.3 to be particularly important, as it specifies two compared variants of the use of no-till cultivation systems, while the first described system corresponded to standard procedures and the second system was based on the use of both machine prototypes. I recommend that it be added here or elsewhere, in which period the experiments were established and what the current climatic or soil conditions were, what characteristics the field on which the experiments were established had (slope of the surface, stones in the soil, etc.). I also think that the parameters of the seed used (purity, germination) should be added, because the results of corn cultivation will certainly depend on them.
Subchapter 2.4 presents a set of measurement methods used in the experiments. Despite the fact that this file is quite extensive, it lacks information that would specify the conditions for setting up experiments, i.e. the characteristics of the field surface (condition of plant residues), the efficiency of the disintegration of plant residues with the stubble cutter prototype, possible malfunctions during sowing with a no-till seeder, affected by plant residues, the variability of the depth of seed placement in the soil, etc. It should also be stated how extensive the experimental areas were, in which spatial system the experiments were repeated (statistics), when the experimental sowings were carried out (first and second year), etc.
Chapter 3 – Results is focused on the presentation of production characteristics when growing maize based on both tested systems (classic cultivation methods and technological methods based on the use of new machine prototypes). If I understand correctly, the production indicators (maize yields) are about the same for both systems tested. However, significant positive differences are found in the evaluation of fuel consumption, where the new stubble cutter prototype shows only approx. 20% fuel consumption compared to the classic stalk shredder. On the contrary, the prototype no-till seeder shows more than twice the fuel consumption compared to the classic no-till seeder. What is the cause of this fact? As I have stated several times before, in Results I am missing the technical indicators of the tested machines (the degree of disintegration of plant residues, the quality of seeding, etc.). I would also expect (taking into account the title of the article) that the parameters of labor productivity, or the consumption of working time, and their comparison between the two monitored systems will be presented here.
In Chapter 4 – Discussion, the authors comment on some of their findings. I won't comment on everyone. I stop at paragraph 2 (line 230). I believe that it should be emphasized more whether the new technology of no-tillage maize cultivation, based on both machine prototypes, provides a sufficient guarantee that in the future farmers will no longer have to burn plant residues on the stubble of their fields. The next paragraph I comment on starts with line 249. Here the authors mention the consumption of working time for both technological systems. However, I consider it a serious shortcoming that in the previous sections of the article it was not stated how specifically these times were determined, whether they were measured or just calculated, and what their sub-values were, which, in summary, make up the values presented on line 250. In short, the consumption of working time should have been presented in chapter 3 Results. I clearly agree with the authors' final statement in the last paragraph of this Discussion chapter (line 274 et seq.) and state that the content of the article, despite some of my comments, supports the opinions presented here.
Chapter 6 – Conclusions briefly declares the most important findings presented in the article (fuel saving, lower tractor performance, cheaper cultivation technology for farmers). I appreciate that the authors of the article objectively recognize that the use of the new no-till seeder prototype can, on the contrary, increase fuel consumption compared to classic means.
At the end of my review, I state that, despite my above-mentioned comments, the article is built on good foundations and that, after making relatively minor adjustments, it can be accepted for publication.
Author Response
First of all, thank you for reviewing our article and sharing your valuable comments with us. Below you can find the arrangements we have made based on your comments and recommendations.

Reviewer 2 Report
The form of the report does not coincide with the template requirements! The methods and methodology are inadequate as well as the figures' quality. I suggest leaving only 3D figures and explaining the figures by numbering the parts of the machines. The methods referred to the pieces of literature (Table 5) but are not explained in the results and discussion part with figures. There is not enough proof (figures with explanation) that the research was conducted. In the article, a conventional stalk shredder is compared with a stubble cutter machine prototype and with a no-till seeder. There is no discussion or explanation about conventional stalk shredder. Moreover, I consider that comparing the stubble cutter machine prototype with the no-till seeder is inadequate as the functions are different. I assume that dealing with the corn stubble should be discussed not only the conditions of your country but how other countries are dealing with it. I suggest resubmitting after increasing the quality of the manuscript.
Author Response

(The authors gave the same response as above.)

Round 2
Reviewer 2 Report
The quality of the figures is increased and numbered. That is good. I would also require mentioning the numbers in the text so the reader would find it easy to understand what the figures are about.
-There is a gap between the paragraphs all over the text!
- Please, give more information about traditional stubble shredders with clear pictures and working principle as long as you are comparing with it.
-In this study, the stubble cutter machine prototype and the no-till seeder were not compared (The aim of this study is to compare the no-till sowing machine prototype and stalk cutting machine prototype with the classical stalk cutter in terms of fuel consumption and the input/output ratios. ((lines 18-20)). On the contrary, the systems formed by using these machines and serving the same purpose were compared with each other. It would be adequate to explain this in the abstract and in the text otherwise it is too confusing! Why use a no-till seeder to prepare the sowing bed, when it is possible directly in one pass to apply seeds and prepare the sowing bed? Do you mean to use a no-till seeder to prepare the seeding bed and then again use the no-till seeder to seed!? I mean no-till seeder provides two operations simultaneously! there was no point to use a no-till seeder to prepare the seeding bed. The prototype stubble-cutting machine used in the study was manufactured with a cylindrical structure and equipped with cutting blades 1 cm thick and cm wide, on the other hand, the no-till seeder prototype was manufactured as a fertilizer seeder equipped with a bucket-type seed hopper capable of sowing 4 rows (In the abstract). Is the no-till seeder capable of seeding only fertilizers?
-What do you mean by input/output ratios in the abstract?
-Figure 1. Design images of the no-till seeder prototype (1. Fertilizer hopper, 2. Seed hopper, 3. Seed press roller, 4. Sowing coulter with cutting disc, 5. Sowing coulter with cutting disc). 4 and 5 are the same?
- In the introduction part there is no need to mentioning about burning the corn stalk because it is not the main problem you are investigating! Fuel consumption and draft force of machines were the main research that you note in the abstract. Therefore, the introduction should be rewritten. Lines 355-367, 262-285, and241-265 should be written in the introduction.
-The aim of this study is to present no-till seeder and stubble cutter prototypes to maize producers that are affordable for small enterprises (Did you compare the prices of the machines you are comparing?). Another aim is to increase the farmers' options for preparing more economical seedbeds without burning stubble. (The aim is not adequate.)
-The averages were achieved by using a penetrelogger with memory, LSD display and GPS source that can measure in the range of 0-80 cm soil depth and 0-5 MPa to measure rows with 3 replications at different depths after harvest. Please, if you could show the figures as proof the research was provided?
-The abstract is recommended to rewrite as the main target is to reduce fuel consumption not burning crop residues!
-Figure 3. Where is the no-till seeder sowing bed preparation?
Author Response
Dear Reviewer,
Below you can find the arrangements we have made based on your comments and recommendations.
Best redards…
